# Distributed Parameter Estimation in Probabilistic Graphical Models

**Yariv D. Mizrahi**[1]    **Misha Denil**[2]    **Nando de Freitas**[2,3,4]
[1]University of British Columbia, Canada
[2]University of Oxford, United Kingdom
[3]Canadian Institute for Advanced Research
[4]Google DeepMind
`yariv@math.ubc.ca`
`{misha.denil,nando}@cs.ox.ac.uk`

## Abstract

This paper presents foundational theoretical results on distributed parameter estimation for undirected probabilistic graphical models. It introduces a general condition on composite likelihood decompositions of these models which guarantees the global consistency of distributed estimators, provided the local estimators are consistent.

## 1   Introduction

Undirected probabilistic graphical models, also known as Markov Random Fields (MRFs), are a natural framework for modelling in networks, such as sensor networks and social networks [24, 11, 20]. In large-scale domains there is great interest in designing distributed learning algorithms to estimate parameters of these models from data [27, 13, 19]. Designing distributed algorithms in this setting is challenging because the distribution over variables in an MRF depends on the global structure of the model.

In this paper we make several theoretical contributions to the design of algorithms for distributed parameter estimation in MRFs by showing how the recent works of Liu and Ihler [13] and of Mizrahi *et al.* [19] can both be seen as special cases of *distributed composite likelihood*. Casting these two works in a common framework allows us to transfer results between them, strengthening the results of both works.

Mizrahi *et al.* introduced a theoretical result, known as the *LAP condition*, to show that it is possible to learn MRFs with untied parameters in a fully-parallel but globally consistent manner. Their result led to the construction of a globally consistent estimator, whose cost is linear in the number of cliques as opposed to exponential as in centralised maximum likelihood estimators. While remarkable, their results apply only to a specific factorisation, with the cost of learning being exponential in the size of the factors. While their factors are small for lattice-MRFs and other models of low degree, they can be as large as the original graph for other models, such as fully-observed Boltzmann machines [1]. In this paper, we introduce the *Strong LAP Condition*, which characterises a large class of composite likelihood factorisations for which it is possible to obtain global consistency, provided the local estimators are consistent. This much stronger condition enables us to construct linear and globally consistent distributed estimators for a much wider class of models than Mizrahi *et al.*, including fully-connected Boltzmann machines.

Using our framework we also show how the asymptotic theory of Liu and Ihler applies more generally to distributed composite likelihood estimators. In particular, the Strong LAP Condition provides a sufficient condition to guarantee the validity of a core assumption made in the theory of Liu and Ihler, namely that each local estimate for the parameter of a clique is a consistent estimator of the

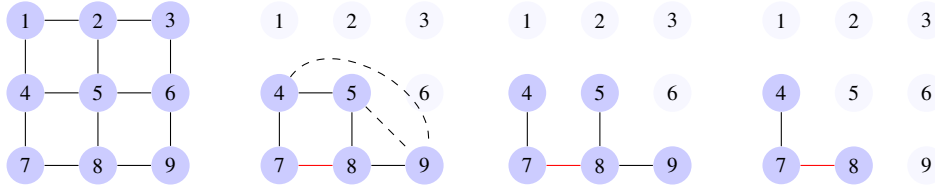

Figure 1: **Left:** A simple 2d-lattice MRF to illustrate our notation. For node $j = 7$ we have $\mathcal{N}(x_j) = \{x_4, x_8\}$. **Centre left:** The 1-neighbourhood of the clique $q = \{x_7, x_8\}$ including additional edges (dashed lines) present in the marginal over the 1-neighbourhood. Factors of this form are used by the LAP algorithm of Mizrahi *et. al.* **Centre right:** The MRF used by our conditional estimator of Section 5 when using the same domain as Mizrahi *et. al.* **Right:** A smaller neighbourhood which we show is also sufficient to estimate the clique parameter of $q$.

corresponding clique parameter in the joint distribution. By applying the Strong LAP Condition to verify the assumption of Liu and Ihler, we are able to import their M-estimation results into the LAP framework directly, bridging the gap between LAP and consensus estimators.

## 2  Background

Our goal is to estimate the $D$-dimensional parameter vector $\boldsymbol{\theta}$ of an MRF with the following *Gibbs density* or *mass function*:

$$p(\mathbf{x} \,|\, \boldsymbol{\theta}) = \frac{1}{Z(\boldsymbol{\theta})} \exp(-\sum_c E(\mathbf{x}_c \,|\, \boldsymbol{\theta}_c)) \tag{1}$$

Here $c \in \mathcal{C}$ is an index over the cliques of an undirected graph $\mathcal{G} = (\mathcal{V}, \mathcal{E})$, $E(\mathbf{x}_c \,|\, \boldsymbol{\theta}_c)$ is known as the *energy* or *Gibbs potential*, and $Z(\boldsymbol{\theta})$ is a normalizing term known as the *partition function*.

When $E(\mathbf{x}_c \,|\, \boldsymbol{\theta}_c) = -\boldsymbol{\theta}_c^T \boldsymbol{\phi}_c(\mathbf{x}_c)$, where $\boldsymbol{\phi}_c(\mathbf{x}_c)$ is a local sufficient statistic derived from the values of the local data vector $\mathbf{x}_c$, this model is known as a *maximum entropy* or *log-linear* model. In this paper we do not restrict ourselves to a specific form for the potentials, leaving them as general functions; we require only that their parameters are identifiable. Throughout this paper we focus on the case where the $x_j$'s are discrete random variables, however generalising our results to the continuous case is straightforward.

The $j$-th node of $\mathcal{G}$ is associated with the random variable $x_j$ for $j = 1, \ldots, M$, and the edge connecting nodes $j$ and $k$ represents the statistical interaction between $x_j$ and $x_k$. By the Hammersley-Clifford Theorem [10], the random vector $\mathbf{x}$ satisfies the Markov property with respect to the graph $\mathcal{G}$, *i.e.*, $p(x_j|\mathbf{x}_{-j}) = p(x_j|\mathbf{x}_{\mathcal{N}(x_j)})$ for all $j$ where $\mathbf{x}_{-j}$ denotes all variables in $\mathbf{x}$ excluding $x_j$, and $\mathbf{x}_{\mathcal{N}(x_j)}$ are the variables in the neighbourhood of node $j$ (variables associated with nodes in $\mathcal{G}$ directly connected to node $j$).

### 2.1  Centralised estimation

The standard approach to parameter estimation in statistics is through maximum likelihood, which chooses parameters $\boldsymbol{\theta}$ by maximising

$$\mathcal{L}^{ML}(\boldsymbol{\theta}) = \prod_{n=1}^{N} p(\mathbf{x}_n \,|\, \boldsymbol{\theta}) \tag{2}$$

(To keep the notation light, we reserve $n$ to index the data samples. In particular, $\mathbf{x}_n$ denotes the $n$-th $|\mathcal{V}|$-dimensional data vector and $x_{mn}$ refers to the $n$-th observation of node $m$.)

This estimator has played a central role in statistics as it has many desirable properties including consistency, efficiency and asymptotic normality. However, applying maximum likelihood estimation to an MRF is generally intractable since computing the value of $\log \mathcal{L}^{ML}$ and its derivative require evaluating the partition function, and an expectation over the model, respectively. Both of these values involve a sum over exponentially many terms.

To surmount this difficulty it is common to approximate $p(\mathbf{x} \mid \boldsymbol{\theta})$ as a product over more tractable terms. This approach is known as *composite likelihood* and leads to an objective of the form

$$\mathcal{L}^{CL}(\boldsymbol{\theta}) = \prod_{n=1}^{N} \prod_{i=1}^{I} f^i(\mathbf{x}_n, \boldsymbol{\theta}^i) \tag{3}$$

where $\boldsymbol{\theta}^i$ denote the (possibly shared) parameters of each composite likelihood factor $f^i$.

Composite likelihood estimators are both well studied and widely applied [6, 14, 12, 7, 16, 2, 22, 4, 21]. In practice the $f^i$ terms are chosen to be easy to compute, and are typically local functions, depending only on some local region of the underlying graph $\mathcal{G}$.

An early and influential variant of composite likelihood is *pseudo-likelihood* (PL) [3], where $f^i(\mathbf{x}, \boldsymbol{\theta}^i)$ is chosen to be the conditional distribution of $x_i$ given its neighbours,

$$\mathcal{L}^{PL}(\boldsymbol{\theta}) = \prod_{n=1}^{N} \prod_{m=1}^{M} p(x_{mn} \mid \mathbf{x}_{\mathcal{N}(x_m)n}, \boldsymbol{\theta}^m) \tag{4}$$

Since the joint distribution has a Markov structure with respect to the graph $\mathcal{G}$, the conditional distribution for $x_m$ depends only on its neighbours, namely $\mathbf{x}_{\mathcal{N}(x_m)}$. In general more statistically efficient composite likelihood estimators can be obtained by blocking, *i.e.* choosing the $f^i(\mathbf{x}, \boldsymbol{\theta}^i)$ to be conditional or marginal likelihoods over blocks of variables, which may be allowed to overlap.

Composite likelihood estimators are often divided into conditional and marginal variants, depending on whether the $f^i(\mathbf{x}, \boldsymbol{\theta}^i)$ are formed from conditional or marginal likelihoods. In machine learning the conditional variant is quite popular [12, 7, 16, 15, 4] while the marginal variant has received less attention. In statistics, both the marginal and conditional variants of composite likelihood are well studied (see the comprehensive review of Varin *et. al.* [26]).

An unfortunate difficulty with composite likelihood is that the estimators cannot be computed in parallel, since elements of $\boldsymbol{\theta}$ are often shared between the different factors. For a fixed value of $\boldsymbol{\theta}$ the terms of $\log \mathcal{L}^{CL}$ decouple over data and over blocks of the decomposition; however, if $\boldsymbol{\theta}$ is not fixed then the terms remain coupled.

## 2.2 Consensus estimation

Seeking greater parallelism, researchers have investigated methods for decoupling the sub-problems in composite likelihood. This leads to the class of *consensus estimators*, which perform parameter estimation independently in each composite likelihood factor. This approach results in parameters that are shared between factors being estimated multiple times, and a final consensus step is required to force agreement between the solutions from separate sub-problems [27, 13].

Centralised estimators enforce sub-problem agreement throughout the estimation process, requiring many rounds of communication in a distributed setting. Consensus estimators allow sub-problems to disagree during optimisation, enforcing agreement as a post-processing step which requires only a single round of communication.

Liu and Ihler [13] approach distributed composite likelihood by optimising each term separately

$$\hat{\boldsymbol{\theta}}_{\beta_i}^i = \arg\max_{\boldsymbol{\theta}_{\beta_i}} \left( \prod_{n=1}^{N} f^i(\mathbf{x}_{\mathcal{A}^i,n}, \boldsymbol{\theta}_{\beta_i}) \right) \tag{5}$$

where $\mathcal{A}^i$ denotes the group of variables associated with block $i$, and $\boldsymbol{\theta}_{\beta_i}$ is the corresponding set of parameters. In this setting the sets $\beta_i \subseteq \mathcal{V}$ are allowed to overlap, but the optimisations are carried out independently, so multiple estimates for overlapping parameters are obtained. Following Liu and Ihler we have used the notation $\boldsymbol{\theta}^i = \boldsymbol{\theta}_{\beta_i}$ to make this interdependence between factors explicit.

The analysis of this setting proceeds by embedding each local estimator $\hat{\boldsymbol{\theta}}_{\beta_i}^i$ into a degenerate estimator $\hat{\boldsymbol{\theta}}^i$ for the global parameter vector $\boldsymbol{\theta}$ by setting $\hat{\boldsymbol{\theta}}_c^i = 0$ for $c \notin \beta_i$. The degenerate estimators are combined into a single non-degenerate global estimate using different consensus operators, *e.g.* weighted averages of the $\hat{\boldsymbol{\theta}}^i$.

The analysis of Liu and Ihler assumes that for each sub-problem $i$ and for each $c \in \beta_i$

$$(\hat{\boldsymbol{\theta}}^i_{\beta_i})_c \xrightarrow{p} \boldsymbol{\theta}_c \tag{6}$$

*i.e.*, each local estimate for the parameter of clique $c$ is a consistent estimator of the corresponding clique parameter in the joint distribution. This assumption does not hold in general, and one of the contributions of this work is to give a general condition under which this assumption holds.

The analysis of Liu and Ihler [13] considers the case where the local estimators in Equation 5 are arbitrary $M$-estimators [25], however their experiments address only the case of pseudo-likelihood. In Section 5 we prove that the factorisation used by pseudo-likelihood satisfies Equation 6, explaining the good results in their experiments.

## 2.3 Distributed estimation

Consensus estimation dramatically increases the parallelism of composite likelihood estimates by relaxing the requirements on enforcing agreement between coupled sub-problems. Recently Mizrahi *et. al.* [19] have shown that if the composite likelihood factorisation is constructed correctly then consistent parameter estimates can be obtained without requiring a consensus step.

In the LAP algorithm of Mizrahi *et al.* [19] the domain of each composite likelihood factor (which they call the *auxiliary MRF*) is constructed by surrounding each maximal clique $q$ with the variables in its *1-neighbourhood*

$$\mathcal{A}_q = \bigcup_{c \cap q \neq \emptyset} c$$

which contains all of the variables of $q$ itself as well as the variables with at least one neighbour in $q$; see Figure 1 for an example. For MRFs of low degree the sets $\mathcal{A}_q$ are small, and consequently maximum likelihood estimates for parameters of MRFs over these sets can be obtained efficiently. The parametric form of each factor in LAP is chosen to coincide with the marginal distribution over $\mathcal{A}_q$.

The factorisation of Mizrahi *et al.* is essentially the same as in Equation 5, but the domain of each term is carefully selected, and the LAP theorems are proved only for the case where $f^i(\mathbf{x}_{\mathcal{A}_q}, \boldsymbol{\theta}_{\beta_q}) = p(\mathbf{x}_{\mathcal{A}_q}, \boldsymbol{\theta}_{\beta_q})$.

As in consensus estimation, parameter estimation in LAP is performed separately and in parallel for each term; however, agreement between sub-problems is handled differently. Instead of combining parameter estimates from different sub-problems, LAP designates a specific sub-problem as authoritative for each parameter (in particular the sub-problem with domain $\mathcal{A}_q$ is authoritative for the parameter $\boldsymbol{\theta}_q$). The global solution is constructed by collecting parameters from each sub-problem for which it is authoritative and discarding the rest.

In order to obtain consistency for LAP, Mizrahi *et al.* [19] assume that both the joint distribution and each composite likelihood factor are parametrised using normalized potentials.

**Definition 1.** *A Gibbs potential $E(\mathbf{x}_c|\boldsymbol{\theta}_c)$ is said to be normalised with respect to zero if $E(\mathbf{x}_c|\boldsymbol{\theta}_c) = 0$ whenever there exists $t \in c$ such that $\mathbf{x}_t = 0$.*

A perhaps under-appreciated existence and uniqueness theorem [9, 5] for MRFs states that there exists one and only one potential normalized with respect to zero corresponding to a Gibbs distribution. This result ensures a one to one correspondence between Gibbs distributions and normalised potential representations of an MRF.

The consistency of LAP relies on the following observation. Suppose we have a Gibbs distribution $p(\mathbf{x}_\mathcal{V} | \boldsymbol{\theta})$ that factors according to the clique system $\mathcal{C}$, and suppose that the parametrisation is chosen so that the potentials are normalised with respect to zero. For a particular clique of interest $q$, the marginal over $\mathbf{x}_{\mathcal{A}_q}$ can be written as follows (see Appendix A for a detailed derivation)

$$p(\mathbf{x}_{\mathcal{A}_q} | \boldsymbol{\theta}) = \frac{1}{Z(\boldsymbol{\theta})} \exp(-E(\mathbf{x}_q | \boldsymbol{\theta}_q) - \sum_{c \in \mathcal{C}_q \setminus \{q\}} E(\mathbf{x}_c | \boldsymbol{\theta}_{\mathcal{V} \setminus q})) \tag{7}$$

where $\mathcal{C}_q$ denotes the clique system of the marginal, which in general includes cliques not present in the joint. The same distribution can also be written in terms of different parameters $\boldsymbol{\alpha}$

$$p(\mathbf{x}_{\mathcal{A}_q} \mid \boldsymbol{\alpha}) = \frac{1}{Z(\boldsymbol{\alpha})} \exp(-E(\mathbf{x}_q \mid \boldsymbol{\alpha}_q) - \sum_{c \in \mathcal{C}_q \setminus \{q\}} E(\mathbf{x}_c \mid \boldsymbol{\alpha}_c)) \tag{8}$$

which are also assumed to be normalised with respect to zero. As shown in Mizrahi *et. al.* [19], the uniqueness of normalised potentials can be used to obtain the following result.

**Proposition 2** (**LAP argument** [19]). *If the parametrisations of $p(\mathbf{x}_{\mathcal{V}} \mid \boldsymbol{\theta})$ and $p(\mathbf{x}_{\mathcal{A}_q} \mid \boldsymbol{\alpha})$ are chosen to be normalized with respect to zero, and if the parameters are identifiable with respect to the potentials, then $\boldsymbol{\theta}_q = \boldsymbol{\alpha}_q$.*

This proposition enables Mizrahi *et. al.* [19] to obtain consistency for LAP under the standard smoothness and identifiability assumptions for MRFs [8].

# 3 Contributions of this paper

The strength of the results of Mizrahi *et al.* [19] is to show that it is possible to perform parameter estimation in a completely distributed way without sacrificing global consistency. They prove that through careful design of a composite likelihood factorisation it is possible to obtain estimates for each parameter of the joint distribution in isolation, without requiring even a final consensus step to enforce sub-problem agreement. Their weakness is that the LAP algorithm is very restrictive, requiring a specific composite likelihood factorisation.

The strength of the results of Liu and Ihler [13] is that they apply in a very general setting (arbitrary $M$-estimators) and make no assumptions about the underlying structure of the MRF. On the other hand they assume the convergence in Equation 6, and do not characterise the conditions under which this assumption holds.

The key to unifying these works is to notice that the specific decomposition used in LAP is chosen essentially to ensure the convergence of Equation 6. This leads to our development of the Strong LAP Condition and an associated Strong LAP Argument, which is a drop in replacement for the LAP argument of Mizrahi *et al.* and holds for a much larger range of composite likelihood factorisations than their original proof allows.

Since the purpose of the Strong LAP Condition is to guarantee the convergence of Equation 6, we are able to import the results of Liu and Ihler [13] into the LAP framework directly, bridging the gap between LAP and consensus estimators. The same Strong LAP Condition also provides the necessary convergence guarantee for the results of Liu and Ihler to apply.

Finally we show how the Strong LAP Condition can lead to the development of new estimators, by developing a new distributed estimator which subsumes the distributed pseudo-likelihood and gives estimates that are both consistent and asymptotically normal.

# 4 Strong LAP argument

In this section we present the Strong LAP Condition, which provides a general condition under which the convergence of Equation 6 holds. This turns out to be intimately connected to the structure of the underlying graph.

**Definition 3** (Relative Path Connectivity). *Let $\mathcal{G} = (\mathcal{V}, \mathcal{E})$ be an undirected graph, and let $\mathcal{A}$ be a given subset of $\mathcal{V}$. We say that two nodes $i, j \in \mathcal{A}$ are* path connected with respect to $\mathcal{V} \setminus \mathcal{A}$ *if there exists a path $\mathcal{P} = \{i, s_1, s_2, \ldots, s_n, j\} \neq \{i, j\}$ with none of the $s_k \in \mathcal{A}$. Otherwise, we say that $i, j$ are* path disconnected with respect to $\mathcal{V} \setminus \mathcal{A}$.

For a given $\mathcal{A} \subseteq \mathcal{V}$ we partition the clique system of $\mathcal{G}$ into two parts, $\mathcal{C}_{\mathcal{A}}^{in}$ that contains all of the cliques that are a subset of $\mathcal{A}$, and $\mathcal{C}_{\mathcal{A}}^{out} = \mathcal{C} \setminus \mathcal{C}_{\mathcal{A}}^{in}$ that contains the remaining cliques of $\mathcal{G}$. Using this notation we can write the marginal distribution over $\mathbf{x}_{\mathcal{A}}$ as

$$p(\mathbf{x}_{\mathcal{A}} \mid \boldsymbol{\theta}) = \frac{1}{Z(\boldsymbol{\theta})} \exp(-\sum_{c \in \mathcal{C}_{\mathcal{A}}^{in}} E(\mathbf{x}_c \mid \boldsymbol{\theta}_c)) \sum_{\mathbf{x}_{\mathcal{V} \setminus \mathcal{A}}} \exp(-\sum_{c \in \mathcal{C}_{\mathcal{A}}^{out}} E(\mathbf{x}_c \mid \boldsymbol{\theta}_c)) \tag{9}$$

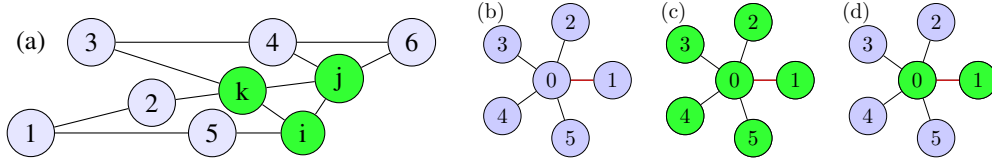

Figure 2: **(a)** Illustrating the concept of relative path connectivity. Here, $\mathcal{A} = \{i, j, k\}$. While $(k, j)$ are path connected via $\{3, 4\}$ and $(k, i)$ are path connected via $\{2, 1, 5\}$, the pair $(i, j)$ are path disconnected with respect to $\mathcal{V} \setminus \mathcal{A}$. **(b)-(d)** Illustrating the difference between LAP and Strong LAP. (b) Shows a star graph with $q$ highlighted. (c) Shows $\mathcal{A}_q$ required by LAP. (d) Shows an alternative neighbourhood allowed by Strong LAP. Thus, if the root node is a response variable and the leafs are covariates, Strong LAP states we can estimate each parameter separately and consistently.

Up to a normalisation constant, $\sum_{\mathbf{x}_{\mathcal{V} \setminus \mathcal{A}}} \exp(-\sum_{c \in \mathcal{C}_{\mathcal{A}}^{out}} E(\mathbf{x}_c \,|\, \boldsymbol{\theta}_c))$ induces a Gibbs density (and therefore an MRF) on $\mathcal{A}$, which we refer to as the *induced MRF*. (For example, as illustrated in Figure 1 centre-left, the induced MRF involves all the cliques over the nodes 4, 5 and 9.) By the Hammersley-Clifford theorem this MRF has a corresponding graph which we refer to as the *induced graph* and denote $\mathcal{G}_{\mathcal{A}}$. Note that the induced graph does not have the same structure as the marginal, it contains only edges which are created by summing over $\mathbf{x}_{\mathcal{V} \setminus \mathcal{A}}$.

**Remark 4.** *To work in the general case, we assume throughout that that if an MRF contains the path $\{i, j, k\}$ then summing over $j$ creates the edge $(i, k)$ in the marginal.*

**Proposition 5.** *Let $\mathcal{A}$ be a subset of $\mathcal{V}$, and let $i, j \in \mathcal{A}$. The edge $(i, j)$ exists in the induced graph $\mathcal{G}_{\mathcal{A}}$ if and only if $i$ and $j$ are path connected with respect to $\mathcal{V} \setminus \mathcal{A}$.*

*Proof.* If $i$ and $j$ are path connected then there is a path $\mathcal{P} = \{i, s_1, s_2, \ldots, s_n, j\} \neq \{i, j\}$ with none of the $s_k \in \mathcal{A}$. Summing over $s_k$ forms an edge $(s_{k-1}, s_{k+1})$. By induction, summing over $s_1, \ldots, s_n$ forms the edge $(i, j)$.

If $i$ and $j$ are path disconnected with respect to $\mathcal{V} \setminus \mathcal{A}$ then summing over any $s \in \mathcal{V} \setminus \mathcal{A}$ cannot form the edge $(i, j)$ or $i$ and $j$ would be path connected through the path $\{i, s, j\}$. By induction, if the edge $(i, j)$ is formed by summing over $s_1, \ldots, s_n$ this implies that $i$ and $j$ are path connected via $\{i, s_1, \ldots, s_n, j\}$, contradicting the assumption. $\square$

**Corollary 6.** *$\mathcal{B} \subseteq \mathcal{A}$ is a clique in the induced graph $\mathcal{G}_{\mathcal{A}}$ if and only if all pairs of nodes in $\mathcal{B}$ are path connected with respect to $\mathcal{V} \setminus \mathcal{A}$.*

**Definition 7** (Strong LAP condition). *Let $\mathcal{G} = (\mathcal{V}, \mathcal{E})$ be an undirected graph and let $q \in \mathcal{C}$ be a clique of interest. We say that a set $\mathcal{A}$ such that $q \subseteq \mathcal{A} \subseteq \mathcal{V}$ satisfies the strong LAP condition for $q$ if there exist $i, j \in q$ such that $i$ and $j$ are path-disconnected with respect to $\mathcal{V} \setminus \mathcal{A}$.*

**Proposition 8.** *Let $\mathcal{G} = (\mathcal{V}, \mathcal{E})$ be an undirected graph and let $q \in \mathcal{C}$ be a clique of interest. If $\mathcal{A}_q$ satisfies the Strong LAP condition for $q$ then the joint distribution $p(\mathbf{x}_{\mathcal{V}} \,|\, \boldsymbol{\theta})$ and the marginal $p(\mathbf{x}_{\mathcal{A}_q} \,|\, \boldsymbol{\theta})$ share the same normalised potential for $q$.*

*Proof.* If $\mathcal{A}_q$ satisfies the Strong LAP Condition for $q$ then by Corollary 6 the induced MRF contains no potential for $q$. Inspection of Equation 9 reveals that the same $E(\mathbf{x}_q \,|\, \boldsymbol{\theta}_q)$ appears as a potential in both the marginal and the joint distributions. The result follows by uniqueness of the normalised potential representation. $\square$

We now restrict our attention to a set $\mathcal{A}_q$ which satisfies the Strong LAP Condition for a clique of interest $q$. The marginal over $p(\mathbf{x}_{\mathcal{A}_q} \,|\, \boldsymbol{\theta})$ can be written as in Equation 9 in terms of $\boldsymbol{\theta}$, or in terms of auxiliary parameters $\boldsymbol{\alpha}$

$$p(\mathbf{x}_{\mathcal{A}_q} \,|\, \boldsymbol{\alpha}) = \frac{1}{Z(\boldsymbol{\alpha})} \exp(- \sum_{c \in \mathcal{C}_q} E(\mathbf{x}_c \,|\, \boldsymbol{\alpha}_c)) \tag{10}$$

Where $\mathcal{C}_q$ is the clique system over the marginal. We will assume both parametrisations are normalised with respect to zero.

**Theorem 9** (Strong LAP Argument). *Let $q$ be a clique in $\mathcal{G}$ and let $q \subseteq \mathcal{A}_q \subseteq \mathcal{V}$. Suppose $p(\mathbf{x}_{\mathcal{V}} \,|\, \boldsymbol{\theta})$ and $p(\mathbf{x}_{\mathcal{A}_q} \,|\, \boldsymbol{\alpha})$ are parametrised so that their potentials are normalised with respect to zero and the parameters are identifiable with respect to the potentials. If $\mathcal{A}_q$ satisfies the Strong LAP Condition for $q$ then $\boldsymbol{\theta}_q = \boldsymbol{\alpha}_q$.*

*Proof.* From Proposition 8 we know that $p(\mathbf{x}_{\mathcal{V}} \mid \boldsymbol{\theta})$ and $p(\mathbf{x}_{\mathcal{A}_q} \mid \boldsymbol{\theta})$ share the same clique potential for $q$. Alternatively we can write the marginal distribution as in Equation 10 in terms of auxiliary variables $\boldsymbol{\alpha}$. By uniqueness, both parametrisations must have the same normalised potentials.

Since the potentials are equal, we can match terms between the two parametrisations. In particular since $E(\mathbf{x}_q \mid \boldsymbol{\theta}_q) = E(\mathbf{x}_q \mid \boldsymbol{\alpha}_q)$ we see that $\boldsymbol{\theta}_q = \boldsymbol{\alpha}_q$ by identifiability. □

### 4.1 Efficiency and the choice of decomposition

Theorem 9 implies that distributed composite likelihood is consistent for a wide class of decompositions of the joint distribution; however it does not address the issue of statistical efficiency.

This question has been studied empirically in the work of Meng *et. al.* [17, 18], who introduce a distributed algorithm for Gaussian random fields and consider neighbourhoods of different sizes. Meng *et. al.* find the larger neighbourhoods produce better empirical results and the following theorem confirms this observation.

**Theorem 10.** *Let $\mathcal{A}$ be set of nodes which satisfies the Strong LAP Condition for $q$. Let $\hat{\theta}_{\mathcal{A}}$ be the ML parameter estimate of the marginal over $\mathcal{A}$. If $\mathcal{B}$ is a superset of $\mathcal{A}$, and $\hat{\theta}_{\mathcal{B}}$ is the ML parameter estimate of the marginal over $\mathcal{B}$. Then (asymptotically):*

$$|\theta_q - (\hat{\theta}_{\mathcal{B}})_q| \le |\theta_q - (\hat{\theta}_{\mathcal{A}})_q|.$$

*Proof.* Suppose that $|\theta_q - (\hat{\theta}_{\mathcal{B}})_q| > |\theta_q - (\hat{\theta}_{\mathcal{A}})_q|$. Then the estimates $\hat{\theta}_{\mathcal{A}}$ over the various subsets $\mathcal{A}$ of $\mathcal{B}$ improve upon the ML estimates of the marginal on $\mathcal{B}$. This contradicts the Cramer-Rao lower bound achieved by the ML estimate of the marginal on $\mathcal{B}$. □

In general the choice of decomposition implies a trade-off in computational and statistical efficiency. Larger factors are preferable from a statistical efficiency standpoint, but increase computation and decrease the degree of parallelism.

## 5 Conditional LAP

The Strong LAP Argument tells us that if we construct composite likelihood factors using marginal distributions over domains that satisfy the Strong LAP Condition then the LAP algorithm of Mizrahi *et. al.* [19] remains consistent. In this section we show that more can be achieved.

Once we have satisfied the Strong LAP Condition we know it is acceptable to match parameters between the joint distribution $p(\mathbf{x}_{\mathcal{V}} \mid \boldsymbol{\theta})$ and the auxiliary distribution $p(\mathbf{x}_{\mathcal{A}_q} \mid \boldsymbol{\alpha})$. To obtain a consistent LAP algorithm from this correspondence all that is required is to have a consistent estimate of $\boldsymbol{\alpha}_q$. Mizrahi *et. al.* [19] achieve this by applying maximum likelihood estimation to $p(\mathbf{x}_{\mathcal{A}_q} \mid \boldsymbol{\alpha})$, but any consistent estimator is valid.

We exploit this fact to show how the Strong LAP Argument can be applied to create a consistent conditional LAP algorithm, where conditional estimation is performed in each auxiliary MRF. This allows us to apply the LAP methodology to a broader class of models. For some models, such as large densely connected graphs, we cannot rely on the LAP algorithm of Mizrahi *et. al.* [19]. For example, for a restricted Boltzmann machine (RBM) [23], the 1-neighbourhood of any pairwise clique includes the entire graph. Hence, the complexity of LAP is exponential in the size of $\mathcal{V}$. However, it is linear for conditional LAP, without sacrificing consistency.

**Theorem 11.** *Let $q$ be a clique in $\mathcal{G}$ and let $x_j \in q \subseteq \mathcal{A}_q \subseteq \mathcal{V}$. If $\mathcal{A}_q$ satisfies the Strong LAP Condition for $q$ then $p(\mathbf{x}_{\mathcal{V}} \mid \boldsymbol{\theta})$ and $p(x_j \mid \mathbf{x}_{\mathcal{A}_q \setminus \{x_j\}}, \boldsymbol{\alpha})$ share the same normalised potential for $q$.*

*Proof.* We can write the conditional distribution of $x_j$ given $\mathcal{A}_q \setminus \{x_j\}$ as

$$p(x_j \mid \mathbf{x}_{\mathcal{A}_q \setminus \{x_j\}}, \boldsymbol{\theta}) = \frac{p(\mathbf{x}_{\mathcal{A}_q} \mid \boldsymbol{\theta})}{\sum_{x_j} p(\mathbf{x}_{\mathcal{A}_q} \mid \boldsymbol{\theta})} \tag{11}$$

Both the numerator and the denominator of Equation 11 are Gibbs distributions, and can therefore be expressed in terms of potentials over clique systems.

Since $\mathcal{A}_q$ satisfies the Strong LAP Condition for $q$ we know that $p(\mathbf{x}_{\mathcal{A}_q} \mid \boldsymbol{\theta})$ and $p(\mathbf{x}_{\mathcal{V}} \mid \boldsymbol{\theta})$ have the same potential for $q$. Moreover, the domain of $\sum_{x_j} p(\mathbf{x}_{\mathcal{A}_q} \mid \boldsymbol{\theta})$ does not include $q$, so it cannot contain a potential for $q$. We conclude that the potential for $q$ in $p(x_j \mid \mathbf{x}_{\mathcal{A}_q \setminus \{x_j\}}, \boldsymbol{\theta})$ must be shared with $p(\mathbf{x}_{\mathcal{V}} \mid \boldsymbol{\theta})$. □

**Remark 12.** *There exists a Gibbs representation normalised with respect to zero for* $p(x_j \mid \mathbf{x}_{\mathcal{A}_q \setminus \{x_j\}}, \boldsymbol{\theta})$. *Moreover, the clique potential for $q$ is unique in that representation.*

Existence in the above remark is an immediate result of the the existence of normalized representation both for the numerator and denominator of Equation 11, and the fact that difference of normalised potentials is a normalized potential. For uniqueness, first note that $p(\mathbf{x}_{\mathcal{A}_q} \mid \boldsymbol{\theta}) = p(x_j \mid \mathbf{x}_{\mathcal{A}_q \setminus \{x_j\}}, \boldsymbol{\theta}) p(\mathbf{x}_{\mathcal{A}_q \setminus \{x_j\}}, \boldsymbol{\theta})$ The variable $x_j$ is not part of $p(\mathbf{x}_{\mathcal{A}_q \setminus \{x_j\}}, \boldsymbol{\theta})$ and hence this distribution does not contain the clique $q$. Suppose there were two different normalised representations for the conditional $p(x_j \mid \mathbf{x}_{\mathcal{A}_q \setminus \{x_j\}}, \boldsymbol{\theta})$. This would then imply two normalised representations for the joint, which contradicts the fact that the joint has a unique normalized representation.

We can now proceed as in the original LAP construction from Mizrahi *et al.* [19]. For a clique of interest $q$ we find a set $\mathcal{A}_q$ which satisfies the Strong LAP Condition for $q$. However, instead of creating an auxiliary parametrisation of the marginal we create an auxiliary parametrisation of the conditional in Equation 11.

$$p(x_j \mid \mathbf{x}_{\mathcal{A}_q \setminus \{x_j\}}, \boldsymbol{\alpha}) = \frac{1}{Z_j(\boldsymbol{\alpha})} \exp(- \sum_{c \in \mathcal{C}_{\mathcal{A}_q}} E(\mathbf{x}_c \mid \boldsymbol{\alpha}_c)) \tag{12}$$

From Theorem 11 we know that $E(\mathbf{x}_q \mid \boldsymbol{\alpha}_q) = E(\mathbf{x}_q \mid \boldsymbol{\theta}_q)$. Equality of the parameters is also obtained, provided they are identifiable.

**Corollary 13.** *If $\mathcal{A}_q$ satisfies the Strong LAP Condition for $q$ then any consistent estimator of $\boldsymbol{\alpha}_q$ in* $p(x_j \mid \mathbf{x}_{\mathcal{A}_q \setminus \{x_j\}}, \boldsymbol{\alpha})$ *is also a consistent estimator of $\boldsymbol{\theta}_q$ in $p(\mathbf{x}_{\mathcal{V}} \mid \boldsymbol{\theta})$.*

### 5.1 Connection to distributed pseudo-likelihood and composite likelihood

Theorem 11 tells us that if $\mathcal{A}_q$ satisfies the Strong LAP Condition for $q$ then to estimate $\boldsymbol{\theta}_q$ in $p(\mathbf{x}_{\mathcal{V}} \mid \boldsymbol{\theta})$ it is sufficient to have an estimate of $\boldsymbol{\alpha}_q$ in $p(x_j \mid \mathbf{x}_{\mathcal{A}_q \setminus \{x_j\}}, \boldsymbol{\alpha})$ for any $x_j \in q$. This tells us that it is sufficient to use pseudo-likelihood-like conditional factors, provided that their domains satisfy the Strong LAP Condition. The following remark completes the connection by telling us that the Strong LAP Condition is satisfied by the specific domains used in the pseudo-likelihood factorisation.

**Remark 14.** *Let $q = \{x_1, x_2, .., x_m\}$ be a clique of interest, with 1-neighbourhood $\mathcal{A}_q = q \cup \{\mathcal{N}(x_i)\}_{x_i \in q}$. Then for any $x_j \in q$, the set $q \cup \mathcal{N}(x_j)$ satisfies the Strong LAP Condition for $q$. Moreover, $q \cup \mathcal{N}(x_j)$ satisfies the Strong LAP Condition for all cliques in the graph that contain $x_j$.*

Importantly, to estimate every unary clique potential we need to visit each node in the graph. However, to estimate pairwise clique potentials, visiting all nodes is redundant because the parameters of each pairwise clique are estimated twice. If a parameter is estimated more than once it is reasonable from a statistical standpoint to apply a consensus operator to obtain a single estimate. The theory of Liu and Ihler tells us that the consensus estimates are consistent and asymptotically normal, provided Equation 6 is satisfied. In turn, the Strong LAP Condition guarantees the convergence in Equation 6.

We can go beyond pseudo-likelihood and consider either marginal or conditional factorisations over larger groups of variables. Since the asymptotic results of Liu and Ihler [13] apply to any distributed composite likelihood estimator where the convergence of Equation 6 holds, it follows that any distributed composite likelihood estimator where each factor satisfies the Strong LAP Condition (including LAP and the conditional composite likelihood estimator from Section 5) immediately gains asymptotic normality and variance guarantees as a result of their work and ours.

## 6 Conclusion

We presented foundational theoretical results for distributed composite likelihood. The results provide us with sufficient conditions to apply the results of Liu and Ihler to a broad class of distributed estimators. The theory also led us to the construction of a new globally consistent estimator, whose complexity is linear even for many densely connected graphs. We view extending these results to model selection, tied parameters, models with latent variables, and inference tasks as very important avenues for future research.

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
