[Supplementary Material]

# A Detailed LAP Derivation

This section gives a detailed derivation of Equation 7 from the main body of the paper. We fix a particular clique of interest $q \in \mathcal{C}$ and select a neighbourhood $\mathcal{A}_q$ as in Mizrahi *et al.* [19] to be the union of all cliques of $\mathcal{G}$ that intersect with the clique of interest

$$\mathcal{A}_q = \bigcup_{c \cap q \neq \emptyset} c \ .$$

We are then interested in writing an expression for the marginal $p(\mathbf{x}_{\mathcal{A}_q} \,|\, \boldsymbol{\theta})$ in terms of the joint distribution over the full graph, $p(\mathbf{x}_{\mathcal{V}} \,|\, \boldsymbol{\theta})$. Our choice of $\mathcal{A}_q$ partitions the clique system $\mathcal{C}$ into two parts

$$\mathcal{C}^{in} = \{c \subseteq \mathcal{A}_q \,|\, c \in \mathcal{C}\}$$
$$\mathcal{C}^{out} = \{c \nsubseteq \mathcal{A}_q \,|\, c \in \mathcal{C}\}$$

In particular, note that $q \in \mathcal{C}^{in}$. With the clique system partitioned in this way we can write the marginal as follows

$$p(\mathbf{x}_{\mathcal{A}_q} \,|\, \boldsymbol{\theta}) = \frac{1}{Z(\boldsymbol{\theta})} \sum_{\mathbf{x}_{\mathcal{V} \setminus \mathcal{A}_q}} \exp\left(-\sum_{c \in \mathcal{C}} E(\mathbf{x}_c \,|\, \boldsymbol{\theta}_c)\right)$$

$$= \frac{1}{Z(\boldsymbol{\theta})} \sum_{\mathbf{x}_{\mathcal{V} \setminus \mathcal{A}_q}} \exp\left(-E(\mathbf{x}_q \,|\, \boldsymbol{\theta}_q) - \sum_{c \in \mathcal{C}^{in} \setminus \{q\}} E(\mathbf{x}_c \,|\, \boldsymbol{\theta}_c) - \sum_{c \in \mathcal{C}^{out}} E(\mathbf{x}_c \,|\, \boldsymbol{\theta}_c)\right)$$

$$= \frac{1}{Z(\boldsymbol{\theta})} \exp\left(-E(\mathbf{x}_q \,|\, \boldsymbol{\theta}_q) - \sum_{c \in \mathcal{C}^{in} \setminus \{q\}} E(\mathbf{x}_c \,|\, \boldsymbol{\theta}_c)\right) \sum_{\mathbf{x}_{\mathcal{V} \setminus \mathcal{A}_q}} \exp\left(-\sum_{c \in \mathcal{C}^{out}} E(\mathbf{x}_c \,|\, \boldsymbol{\theta}_c)\right)$$

where the final equality follows from the fact that the first summation contains only cliques whose intersection with $\mathbf{x}_{\mathcal{V} \setminus \mathcal{A}_q}$ is empty.

With some loss of notational precision we can evaluate the sum over the remaining terms

$$p(\mathbf{x}_{\mathcal{A}_q} \,|\, \boldsymbol{\theta}) = \frac{1}{Z(\boldsymbol{\theta})} \exp\left(-E(\mathbf{x}_q \,|\, \boldsymbol{\theta}_q) - \sum_{c \in \mathcal{C}^{in} \setminus \{q\}} E(\mathbf{x}_c \,|\, \boldsymbol{\theta}_c)\right) \exp\left(-\sum_{c \in C_q^{out}} E(\mathbf{x}_c \,|\, \boldsymbol{\theta}_{\mathcal{V} \setminus q})\right)$$

Here we have made two replacements:

1. $\mathcal{C}^{out}$ has been replaced with $\mathcal{C}_q^{out}$, where $\mathcal{C}_q^{out}$ is used to mean whatever clique system arises by summing the final term over $\mathbf{x}_{\mathcal{V} \setminus \mathcal{A}_q}$. Computing the exact structure of $\mathcal{C}_q^{out}$ is in general a hard problem, but for our purposes it suffices to know that $\mathcal{C}_q^{out}$ does not contain $q$, which is clear since all cliques in $\mathcal{C}^{out}$ are disjoint from $q$ by construction.

2. Each potential function in the second term now depends on $\boldsymbol{\theta}_{\mathcal{V} \setminus q}$. Figuring out exactly which cliques depend on which parameters would require us to compute the structure of $\mathcal{C}_q^{out}$; however, we require only that none of the cliques in the second term depend on $\boldsymbol{\theta}_q$, which we know since this term does not contain $\boldsymbol{\theta}_q$.

Finally, by letting $\mathcal{C}_q = (\mathcal{C}^{in} \setminus \{q\}) \cup \mathcal{C}^{out}$ we can write this equation in a more compact way,

$$p(\mathbf{x}_{\mathcal{A}_q} \,|\, \boldsymbol{\theta}) = \frac{1}{Z(\boldsymbol{\theta})} \exp\left(-E(\mathbf{x}_q \,|\, \boldsymbol{\theta}_q) - \sum_{c \in \mathcal{C}_q \setminus \{q\}} E(\mathbf{x}_c \,|\, \boldsymbol{\theta}_c)\right)$$

which corresponds to the form shown in Equation 7.