[Reviews · NeurIPS 2014]

Submitted by Assigned_Reviewer_25

This paper describes a novel, more general, condition for which decentralized parameter estimation on a MRF converges to the consistent global estimator.

This paper is clearly organized and well written. I’m not familiar enough with the literature to make any comment on the magnitude of the theoretical contribution, which seem to explain--rigorously--why the Pseudo-Likelihood method works so well. Technically, the strong LAB condition is general and can work for cliques that are more complicated than the unary partition. However, from distributed algorithms point of few the Pseudo-Likelihood method seems like the best case scenario, only one node and its neighbors. In practice why would you ever use larger sets? And if the PL approach is already being used, does the strong LAB condition “only” provide a new proof for something that people already do, or does it allow you to come up with other partitions? If they exist, the paper would be stronger if it gave examples of other useful partitions to which the strong LAB condition applies, perhaps for better convergence as discussed in 4.1.

Detailed comments:

Check the spelling of centralized throughout the text.

I don’t get the mathematical transformation from line 1 to line 2 of (7), could you give a textual hint or add another intermediate step? (~208)

Remark 4 and Proposition 5 mix “graph language” and “probabilistic language”, specifically summing over edges, and the “induced graph G_A”. In graph language the induced graph would be the induced (sub-)graph by only keeping vertices and edges contained in A. Here it means “induced” by (9), right? I’m not sure how to fix this but coming more from a graph background, reading this passage is confusing.

Summary: This paper describes a novel, more general, condition for which decentralized parameter estimation on a MRF converges to the consistent global estimator.

Submitted by Assigned_Reviewer_27

One way to deal with the computational complexity of maximum likelihood parameter estimation for MRFs is to maximize a composite likelihood, which is related to the likelihood but not equal to it, e.g., pseudo-likelihood. The main result of this paper is the characterization of when an MRF can be decomposed into smaller components where local optimization of the composite likelihood yields a globally consistent estimator of the composite likelihood function. The decomposition is done according to the introduced "strong LAP" condition. The decomposition allows for completely parallel training of the components. Experimental evaluation is not provided, furthermore, providing examples of MRFs where the decomposition has practical benefits could have been helpful for understanding the practical impact and limitations of the decomposition (the same way it was described that the complexity of LAP for restricted Boltzmann machines is exponential, while it is linear for conditional LAP).

Typos:
Line 110 "are and both" -> "are both"
Eq.7: What happened to the summation outside of the \exp?
Line 340: "achieved the by" -> "achieved by"
Summary: + Conditions for maximizing composite likelihoods are provided, allowing for a parallel optimization.
- There is no experimental evaluation, making it hard to evaluate the practical impact of the paper.

Submitted by Assigned_Reviewer_33

Paper Summary:
--------------

The paper focuses on the problem of distributed parameter estimation in MRFs through the use of a composite likelihood function. The authors identify a class of composite likelihood functions for which one can learn the parameters of an MRF in a fully-parallel but globally consistent manner. The authors introduce the Strong LAP Condition characterizing the graph structure of an MRF and show that if the auxiliary MRF A_q corresponding to a clique for interest q satisfies the Strong LAP condition then globally consistent estimates of the parameters corresponding to A_q can be computed locally. This new condition on the structure of the MRF allows the authors to tie together two prior approaches by Liu et al, and Mizrahi et al. for distributed parameter estimation. Moreover, the conditional LAP formulation extends the original LAP algorithm to densely connected graphs.

High-level comments:
-------------------

The paper studies the important problem of distributed parameter estimation in PGMs. I think the techniques put forward in this paper provide an interesting theoretical basis for characterizing families of networks that are amenable to efficient distributed learning and inference algorithms with consistency guarantees. The technical development is sound and the corresponding definitions, prepositions and theorems are clearly stated.

The paper presents some novel ideas as it extends prior work on LAP. More precisely, prior work was using marginal distributions to estimate network parameters, thus, evaluating the partition function, while the current framework shows how parameter estimates can be derived using conditional distributions, and hence, avoiding the computation of the partition function. Finally, the proposed framework unifies two different models previously proposed.

However, the discussion on the differences between LAP and Strong LAP should be improved. Detailed comments on this can be found in the weak points below.

Detailed comments:
-------------------

1) Strong points:
(a) Interesting formulations and sound theoretical analysis.
(b) Unifying framework over two different lines of work. To some extend this work completes is a complement to previous work by Liu et al. by providing the necessary conditions under which the assumptions of Liu et al. hold.
(c) The theoretical framework was used to derive efficient localized parameter estimators. It will be very interesting how these techniques apply to real data, and how they compare with LAP under instances where LAP is also efficient.

2) Weak points:
(a) The comparison between LAP and strong LAP should be improved. Now, Theorem 9 and Proposition 2 seem to state the same thing. Notice that the way they are stated Theorem 9 includes Proposition 2, i.e., normalized potentials and identifiable parameters. So either Theorem 9 or Proposition 2 should be restated as there is no difference between them. In fact theorem 9 seems to be trying to draw a connection between preposition 8 and the previously stated LAP condition but this is done without success.

(b)The discussion in the intro should be extended to give a high-level description of why the Strong LAP condition leads to a more efficient estimator. In fact the reason is the computation of conditional versus marginal distributions. Also, section 5 should be extended with examples. Figure 1 is there but no mention of it is included in section 5.

(c) The statement about the complexity of LAP being exponential in the number of cliques (the same for the centralized estimators) should be that it is exponential in the number of network variables since computing the partition function enumerates all possible assignments of values to these variables. So basically exponential in the clique size but not the number of cliques.

(d) Typos: Some minor typos
Page 1, line 034 allows us transfer -> allows us to tranfer
Page 1, line 039 lead -> led
Page 1, line 040 factorization - > factorization
Page 2, line 086 i.e. -> i.e.,
Page 3, line 160 i.e. that -> i.e., (without that)
Page 3, line 143 optimising - > optimizing
Page 3, line 149 optimisation -> optimization

Summary: The paper presents an interesting theoretical framework for distributed parameter estimation in MRFs. While the analysis is sound the comparison between the proposed work and previous work (especially LAP) should be improved.
Author Feedback
Author rebuttal: We thank all the reviewers for the time and effort they devoted to these very helpful reviews.

Reviewer 25:

You question about why ever use larger neighbourhoods is a very good one. Section 4.1 of the paper shows that larger neighbourhoods are more statistically efficient. There is also a large body of statistical literature comparing composite likelihood against pseudo-likelihood, arguing for larger neighbourhoods. We agree however that there are computational trade-offs in the distributed setting that we must pay attention too. Trading-off storage, communication, statistical and computational efficiency is a topic of vast importance. Theorem 10 makes an inroad into this, but much more could be said in future work.

Let us focus on your second question: “does the strong LAB condition “only” provide a new proof for something that people already do, or does it allow you to come up with other partitions?”
The answer is that if you want to partition your problem in a particular way, the strong LAP theorem tells you whether that is a valid partition. It does not provide a way of comparing different valid partitions. Theorem 10 is an attempt in the latter direction, but it must be emphasized that being able to distinguish valid from non-valid partitions is already of great interest.

Thank you for the detailed comments. We will address these in the final version. In particular, we will add an appendix that shows the derivation of equation (7) in detail.

Reviewer 27:

“Experimental evaluation is not provided, furthermore, providing examples of MRFs where the decomposition has practical benefits could have been helpful for understanding the practical impact and limitations of the decomposition (the same way it was described that the complexity of LAP for restricted Boltzmann machines is exponential, while it is linear for conditional LAP).”

In contrast to LAP, strong LAP is particularly effective when one of the nodes in a pairwise clique has few neighbours. For example, for a star graph (hub and spokes), the LAP argument results in a decomposition where the sub-problems are as large as the original problem. However, the strong LAP argument allows for smaller components in the decomposition; e.g. single pairs (hub and one spoke).

This paper advances theoretical tools for deciding when algorithms are consistent. The paper focuses on this much needed theory and not on empirically comparing estimators.

We are very thankful for your other suggestions and catching typos. As pointed out above, we will provide a longer derivation of equation (7), also used by Mizrahi et al in their ICML paper. It is perhaps notationally cumbersome, but it is an elementary result once spelt out in detail.

Reviewer 33:
We would like to address your suggested weak points first.

(a) The comparison between LAP and strong LAP should be improved.

We agree with this remark. First, we will add a graph example, as outlined to Reviewer 25 above, that shows the power of the strong LAP argument over the LAP argument. We should have emphasized that Proposition 2, proved in Mizrahi et al, requires a decomposition using marginals over 1-neighbourhoods (as in Figure 1, center left), while Theorem 9 requires only that the decomposition satisfy the Strong LAP condition, which is much weaker. We agree that this difference is not clear as currently written, and we will fix this.

Proposition 8 is about matching clique potential functions between the joint and the marginal. This is a needed step toward Theorem 9, which goes further by matching the parameters of these potential functions.

Regarding (b), we use the strong LAP argument to derive the conditional LAP estimator. However, there are two things at work here
1. The domain of the factors in the decomposition (which A_q’s are valid)
2. The estimator used in each sub-problem
Strong LAP addresses only the first issue. Conditional LAP is a particular estimator for a valid partition in accordance with the strong LAP argument. Our inclusion of the star-graph example (comparing marginal estimators using LAP and strong LAP arguments) will help make this more clear.

Regarding (c) on complexity, you are absolutely right. This is what we intended to say, but clearly what we wrote is wrong. We will fix it.

Thank you very much for pointing out several typos and carefully reading our paper.